# Comparison of Drying Techniques for Extraction of Bioactive Compounds from Olive-Tree Materials

**DOI:** 10.3390/foods12142684

**Published:** 2023-07-12

**Authors:** Ana Castillo-Luna, Hristofor Miho, Carlos A. Ledesma-Escobar, Feliciano Priego-Capote

**Affiliations:** 1Department of Analytical Chemistry, Campus of Rabanales, University of Córdoba, 14014 Córdoba, Spain; t72calua@uco.es (A.C.-L.); z32leesc@uco.es (C.A.L.-E.); 2Chemical Institute for Energy and Environment (IQUEMA), Campus of Rabanales, University of Córdoba, 14014 Córdoba, Spain; 3Maimónides Institute Biomedical Research (IMIBIC), Reina Sofía University Hospital, University of Córdoba, 14014 Córdoba, Spain; 4Consortium for Biomedical Research in Frailty & Healthy Ageing, CIBERFES, Carlos III Institute of Health, 28029 Madrid, Spain; 5Department of Agronomy, Maria de Maeztu Unit of Excellence, Campus of Rabanales, University of Cordoba, 14014 Córdoba, Spain; hmiho@uco.es

**Keywords:** phenols, triterpenes, drying, infrared, microwaves, lyophilization

## Abstract

Olive tree vegetal materials are considered a powerful source for the isolation of bioactive compounds—mainly phenols and triterpenic acids. However, the high humidity content of them reduces their preservation and extractability to a liquid solvent. Accordingly, a drying step is crucial to homogenize the material and to obtain an efficient extraction. We studied the influence of the drying process on the extraction efficiency of bioactive compounds from olive vegetal material. For this purpose, we evaluated the effects of four drying processes on the solid–liquid extraction of bioactive compounds from two by-products, olive leaves and pomace, and olive fruits harvested from two cultivars, Alfafara and Koroneiki. Infrared-assisted drying (IAD) was the most suited approach to obtain extracts enriched in oleuropein from leaves (28.5 and 22.2% dry weight in Alfafara and Koroneiki, respectively). In the case of pomace, lyophilization and microwave-assisted drying led to extracts concentrated in oleacein and oleuropein aglycone, whereas IAD and oven-drying led to extracts with enhanced contents of hydroxytyrosol glucoside and hydroxytyrosol, respectively. The drying process considerably affected the chemical composition of extracts obtained from fruits. Changes in the composition of the extracts were explained essentially by the drying process conditions using auxiliary energies, temperature, and time, which promoted chemical alterations and increased the extractability of the compounds. Therefore, the drying protocol should be selected depending on the phenolic content and initial raw material.

## 1. Introduction

The health benefits of virgin olive oil (VOO) are attributed to the balanced composition between major and minor compounds. Among the minor compounds, it is worth mentioning phenols [1] due to the bioactive properties recognized by the EFSA [2]. Previous studies have shown that phenols can be also found in raw materials derived from olive cultivation such as leaves and olive mill waste residues [3,4]. At present, agri-food industry by-products are growing and constitute an environmental problem [5]. For this reason, research is targeted at designing strategies to manage these residues and minimize the pollution load by reusing them in other activities [3].

In the case of olive mill waste residues, the profile of bioactive components depends on the extraction system, which can involve two or three phases by the outputs resulting after extraction [3,6]. A two-phase system allows for obtaining olive oil and a semisolid olive mill waste residue named “alperujo”, whereas the three-phase system generates olive oil, pomace, and mill wastewater. Currently, olive mill wastewater is considered the most polluting by-product of the olive industry due to its high salinity, acidity, and organic content [7]. Thus, the two-phase system is progressively replacing the three-phase approach in the olive oil industry, which is also supported by the production of high-quality olive oils [3]. Moreover, the two-phase system allows for reducing the economic expenses of the olive oil industry [6]. On the other hand, olive leaves have been traditionally burned or crushed. However, the gas emissions contribute to global warming [3], and for this reason, numerous studies have been alternatively focused on leaves’ composition because of their phenolic content. Therefore, olive leaves are considered a rich source of bioactive compounds, and their exploitation would lead to reducing the greenhouse effect [8].

Olive fruits also contain bioactive compounds such as phenols or triterpenes, which are found at high concentrations when the fruit quality is optimum. Thus, olives can be another source of bioactive compounds to be used in the pharmaceutical, nutraceutical, and food industries [3,9].

Before the isolation of bioactive compounds, olive vegetal materials and by-products are frequently dried to avoid fermentation and oxidative transformation. Rahmanian et al. (2015) [10] reviewed different techniques for the dehydration of olive leaves prior to phenolic extraction. They observed that the phenolic content depends on the dehydration method [10]. Malik and Bradford studied the content of some phenols in olive leaves after drying. They established that the best method to keep the phenolic content was drying at room temperature, which allowed full recovery of oleuropein and verbascoside, although it led to the partial loss of luteolin-O-glucoside derivatives. This drying method was compared with oven drying at 60 °C for four hours, which reduced the phenolic content by up to 50% [11]. Erbay et al. analyzed how drying conditions influence the phenolic content of olive fruits. They observed that if the drying temperature is very high and extended for a prolonged time, olive samples could significantly lose their phenols content [12,13].

In another study, Ahmad-Qasem et al. concluded that olive leaf extracts were richer in oleuropein, verbascoside, and luteolin-7-O-glucoside when samples were dried at 120 °C for 12 min as compared to 70 °C for 50 min. They attributed these results to the inactivation of oxidative enzymes at prolonged times. They also verified that hot air at 120 °C produced a low impact on secoiridoids (oleuropein) and flavonoids (luteolin), but it significantly affected anthocyanins. Hence, the relation between time and temperature resulted crucial to minimize the degradation of bioactive products during drying [14]. According to Ahmad-Qasem et al. (2013) [14], lyophilization provided worse results regarding total phenol content in olive leaves versus hot air drying. On the other hand, phenolic variation was studied in olive leaves from four olive cultivars (Chemlali, Chemchali, Zarrazi, and Chetoui varieties) after drying by infrared and blanching at various temperatures. They concluded that the total phenols concentration increased with temperature [15]. In other studies, the authors evaluated sun drying effects in olive leaves to conclude that the longer the drying time is, the lower the phenolic content is [16,17,18]. Finally, Ghanem et al. (2012) [19] found that microwave drying significantly diminished the phenols concentration in lemon and mandarin. Thus, they concluded that the drying time had to be reduced while microwaves’ irradiation power had to be increased to keep the phenolic content [19].

It is worth mentioning that most published studies dealing with the evaluation of drying techniques for the isolation of phenols refer to total phenol content, and few studies have been targeted at specific compounds. Therefore, the main aims of this paper were (i) to test the effect of four drying strategies (oven heating, infrared irradiation, microwave irradiation, and lyophilization) as pretreatments for the isolation of bioactive compounds; (ii) to apply these drying processes to three different vegetal materials (leaves, pomace, and olive fruits) with a variable composition; and (iii) to evaluate the influence of these drying techniques on the preservation of two main families of bioactive compounds (phenols and triterpenes) in these raw materials.

## 2. Materials and Methods

### 2.1. Samples

Two cultivars, Koroneiki and Alfafara, were selected for this study because they provide VOOs with different phenolic profiles according to the study reported by Miho et al. (2021) [20]. Koroneiki provides VOO with predominantly aglycon isomers of oleuropein and ligstroside, while Alfafara VOO tends to be enriched in oleacein and oleocanthal [20]. Olive fruits and leaves were collected on 15 October 2018, when fruits were characterized by an intermediate ripening index (red-purple fruit color). An additional batch of olive fruits was also collected from both cultivars for the extraction of VOOs using an Abencor extraction system (MC2 Ingeniería y Sistemas, Seville, Spain) following the recommendations provided by the manufacturer [21]. Olive pomace was also sampled for inclusion in this research. The cultivars belong to the World Olive Germplasm Bank of Cordoba (WOGB) (CAP-UCO-IFAPA), which is located at the University of Cordoba [22]. Samples were stored at −80 °C until they were processed.

### 2.2. Reagents and Standards

For sample preparation, we used LC-grade methanol, chloroform, acetone, and ethanol from Scharlab (Barcelona, Spain). As an ionization agent, we selected MS-grade formic acid. 2-Propanol and acetonitrile from Fisher Scientific (Madrid, Spain) were used for the preparation of chromatographic mobile phases. Deionized water (18 MΩ cm) was obtained from a Milli-Q water purification system.

Standards of those compounds that are commercially available were used to confirm the identification. Hydroxytyrosol, oleuropein, maslinic acid, oleanolic acid, nuzhenide, and GL3 were purchased from Extrasynthese (Genay, France). Apigenin, luteolin, luteolin-7-O-glucoside, luteolin-7-rutinoside, quercitrin, quercetin-3-glucoside, rhoifolin, rutin, oleacein, oleuropein aglycone, ligstroside aglycone, and oleocanthal were from Sigma-Aldrich (St. Louis, MO, USA). The purity of all of them was greater than 95%.

### 2.3. Drying of Samples

A set of 72 samples (2 cultivars × 3 types of samples × 4 dehydration protocols × 3 biological replicates) was considered in this study, aiming to evaluate the effect of drying processing conditions on the composition of extracts obtained from different matrices. The four dehydration techniques were lyophilization, oven drying, microwave-assisted drying (MAD), and infrared-assisted drying (IAD). Drying was carried out up to constant weight with all techniques. Table 1 lists the conditions required for drying each sample by application of the different techniques. Lyophilization was carried for 24–48 h. Oven drying was completed at 45 °C for 24–48 h. MAD was programmed in 5-minute cycles with irradiation at 90 W and led to a constant weight in 40–80 min, whereas IAD required 2–24 h. The amount of sample processed in each test was 100 g.

Residual humidity was measured in the different raw materials by using the reference AOAC method (1990) [23].

### 2.4. Metabolites Extraction

The dried samples (1 g) were subjected to solid–liquid extraction with 20 mL of extractant composed of 70/10/10/10 (% in volume) ethanol, water, acetone, and chloroform. The extractions were carried out by shaking for 60 min at 900 vibrations/min by a Vibromatic vibrator-shaker (J.P. Selecta, Barcelona, Spain). The liquid phase was filtered with a 0.22 µm porosity filter to collect 200 µL of each extract. The aliquots were stored in the dark at −20 °C until LC–MS analyses. A pool of extracts was prepared for the identification of phenolic compounds.

### 2.5. LC–QTOF MS/MS Analysis

An Agilent 1200 liquid chromatograph (Agilent Technologies, Palo Alto, CA, USA) furnished with a Zorbax Eclipse Plus C18 chromatographic column (1.8 μm particle size, 150 × 3.0 mm i.d., Agilent Technologies) was utilized for the separation of metabolites. Deionized water (phase A) and a mix of acetonitrile and 2-propanol (70:30 *v*/*v*, phase B) were used as mobile phases. Formic acid was used as an ionization agent in both mobile phases at 0.1% (*v*/*v*). The LC pump was programmed at 0.25 mL/min, and the sample injection volume was 2 µL. The analytical column was kept at 30 °C. The elution gradient was as follows: phase B was maintained at 4% as the initial composition, then increased to 25% from min 0 to 1, from 25% to 40% from min 1 to 6, up to 60% from min 6 to 8, and from 60 to 100% from min 8 to 10. This last composition was maintained for 10 min with the purpose of ensuring the metabolites elution and column cleaning. A post-time of 13 min was used to equilibrate the initial conditions and prepare the subsequent analysis.

An Agilent 6540 quadrupole-time of flight (QTOF) high-resolution hybrid detector (Agilent Technologies, Santa Clara, CA, USA) was used for determination of metabolites. The electrospray ionization (ESI) source parameters operating in negative ionization mode were as follows: nebulizer gas at 45 psi, flow rate of 10 mL, and temperature of the N_2_ as drying gas of 325 °C. The capillary voltage was set at 3500 V, while the Q1, skimmer, and octapole voltages were fixed at 130, 65, and 750 V, respectively. Data were acquired in centroid mode in the extended dynamic range (2 GHz). A full scan was carried out at 6 spectra per second within the *m*/*z* range of 100–1100, with subsequent activation of the three most intense precursor ions by MS/MS using a collision energy of 20 and 40 eV at 3 spectra per second within the *m*/*z* range 60–1100. After acquisition of the first MS/MS spectrum, an exclusion window of 0.75 min was programmed with the purpose of avoiding repetitive fragmentation of the most abundant precursor ions. Constant internal calibration was carried out during the analysis with the use of signals at *m*/*z* 112.9856 (trifluoroacetic acid anion) and *m*/*z* 1033.9881 (hexakis(1H, 1H, 3H-tetrafluoropropoxy)phosphazine, HP-921).

The reference standard oleuropein was used for relative quantitation since this phenol is the main precursor of all secoiridoids. A calibration model was prepared in the range of concentrations 1–20 μg/g. The concentration of identified compounds was expressed as oleuropein equivalents (mg/kg, dry weight).

### 2.6. Data Processing and Statistical Analysis

The data obtained by LC-QTOF M/MS were processed by MassHunter Profinder (version B10.00; Agilent Technologies, Santa Clara, CA, USA). The application of this package allowed the extraction of potential molecular features (MFs). According to the recursive extraction algorithm, all ions exceeding 1000 counts were considered for extraction. Moreover, the isotopic distribution to consider a molecular feature as valid should be defined by two or more ions (with a peak spacing tolerance of *m*/*z* 0.0025, plus 10.0 ppm in mass accuracy). Extraction of MFs considered tentative [M−H]^−^ ions and the formation of formic acid and chloride adducts (HCOO–, Cl–). Neutral loss of water molecules by dehydration was also evaluated. Molecular features were identified based on their retention time (RT), peak intensity, and accurate mass. Subsequently, the recursion step ensured the accurate integration of these entities in each analysis.

The software MassHunter Qualitative v. 10.0 was utilized for the targeted extraction of MS/MS information related to the monitored MF in the whole dataset. This process was carried out for identification once all MFs were extracted and aligned. This information was used for absolute identification using commercially available standards considering both the MS/MS spectra and the retention time. When no commercial standards were available, tentative identification of metabolites was achieved by searching in the MassBank of North America (MoNA; https://mona.fiehnlab.ucdavis.edu/spectra/search, accessed on 31 May 2023) database and others belonging to the research group. Ultimately, the compounds that were not reported in the databases or available as commercial standards were identified by examining the neutral mass losses in conjunction with the distinctive fragmentation patterns of their derivatives, which were obtained from commercially available standards.

Once the signal alignment was completed, the obtained chromatographic peaks were integrated to obtain a clean data matrix. R free software (version 4.2.3, http://www.r-project.org/, accessed on 5 June 2023) was used for further processing and statistical analysis. Statistical analysis included the Kruskal–Wallis test (95% confidence interval) and pairwise comparisons (Wilcox test) to identify significant differences in the relative concentration of identified compounds. Principal component analysis (PCA) was used to identify discrimination patterns among samples. PCA was performed with the mixOmics package by selecting eight components and centralized data and without scaling [24].

## 3. Results and Discussion

### 3.1. Characterization of Bioactive Extracts from Selected Raw Materials

The first step was the characterization of bioactive compounds found in the extracts prepared from olive fruits, pomace, and leaves. For this purpose, we prepared three pools by mixing aliquots of the extracts from each raw material to consider tentative alterations occurring due to the drying procedures. The monitored compounds were phenols and precursors but also triterpenes, particularly triterpenic acids, which are the most abundant in these samples [25,26,27]. A total of 33 bioactive compounds were identified (18 secoiridoids, 5 simple phenols, 8 flavonoids, and 2 triterpenes). The identification parameters are summarized in Appendix A.

Two cultivars, Koroneiki and Alfafara, were selected for this study due to their different phenolic profiles in virgin olive oil and pomace [20]. This difference was revealed in the olive fruit extracts. Extracts from Alfafara fruits contained oleacein as one of the most abundant compounds, while Koroneiki extracts were enriched in oleuropein aglycone. Fruit extracts of both cultivars were dominated by loganin, elenolic acid, and the dialdehydic form of decarboxymethyl elenolic acid. Complementarily, Alfafara fruit extracts also presented hydroxytyrosol glucoside and oleoside-11-methyl-ester as dominating compounds, while Koroneiki extracts reported maslinic acid and luteolin among the most abundant components.

Regarding olive pomace, the extracts from both cultivars highlighted an abundant content of oleacein, hydroxytyrosol glucoside, loganin, and verbascoside. Differentially, Alfafara pomace extract also contained oleoside-11-methyl ester and GL3, while Koroneiki pomace presented a remarkable concentration of hydroxyloganin and oleuropein aglycone. The analysis of extracts from leaves showed a quite similar composition in both cultivars; the two extracts showed a predominance of oleuropein, but also of oleuropein aglycone and oleacein. Furthermore, extracts from leaves also contained a high relative concentration of luteolin-7-O-glucoside, hydroxytyrosol glucoside, and verbascoside.

Figure 1 shows a comparison of the relative concentration of the main bioactive families of compounds found in extracts from the three raw materials. These were previously subjected to lyophilization as the reference drying method [28]. Leaves showed the highest content of most families except for simple phenols, which were more concentrated in pomace. In general terms, the extracts from Alfafara samples were more enriched in these compounds than those obtained from Koroneiki.

The variability of extracts’ composition obtained from olive fruits, leaves, and pomace was evaluated by principal component analysis (PCA) to identify the main discrimination pattern: cultivar, drying treatment, or raw material. The 3D PCA scores plot shows that the principal variability source (around 34.9% for PC1-PC3, Figure 2) was the raw material, which is clearly supported by the composition previously described. For this reason, the dataset was divided into three subsets including results for the three independent vegetal matrices. The three individual PCAs clearly showed that the drying method was the main factor explaining the variability in the composition of the extracts (Appendix A). In the three raw materials, lyophilized samples were partially different from those obtained by heating or with the assistance of auxiliary energies, microwaves, or infrared energy. For this reason, an independent evaluation of the influence of drying treatments for each raw material was carried out. Thus, the influence of the drying conditions was studied pertaining to the main bioactive components of each material.

Concerning the drying procedures (Table 1), leaves were the raw material that was dried faster, followed by pomace and olive fruits. The humidity content was 47% in olive leaves, 51% in fruits, and 58% in pomace. MAD was the fastest drying approach as it required 40 min for leaves, 50 min for pomace, and 50–80 min in fruits. It is worth mentioning that irradiation was carried out at a low power, 90 W, to avoid overheating. A longer drying time was required for Alfafara, with a fruit size bigger than Koroneiki. For IAD, we found relevant differences in processing time, where leaves were dried after 2 h, pomace after 12 h, and fruits required 24 h. Lyophilization and oven drying were quite similar in efficiency since leaves were dried after 24 h, while pomace and fruits required 48 h to complete the drying process.

### 3.2. Influence of Drying on the Extraction of Bioactive Compounds from Olive Leaves

Oleuropein is the most concentrated phenol in leaves extracted after lyophilization, assuming a 28.5% and 22.2% concentration of dry weight in Alfafara and Koroneiki, respectively. These results support the elevated contents of secoiridoids in leaves. When olive leaves were oven-dried, we observed a relevant decrease in oleuropein as compared to lyophilization (15.35% in Alfafara and 7.76% in Koroneiki). The same behavior was found in the extracts obtained after MAD, which also provided a reduction in oleuropein concentration. On the other hand, we observed a higher concentration of oleuropein in extracts from leaves dried by IAD as compared to lyophilization (Figure 3). This is explained by the fact that IAD does not involve direct heating of the sample, thus allowing its preservation [15]. According to several studies, subsequent thawing of the leaves produces a sharp reduction in oleuropein content because of cell membrane breakage by ice crystals. This alteration causes activation of degrading enzymes and, consequently, could explain the observed decrease in oleuropein concentration [14,29].

Complementarily, oleuropein aglycone and oleacein were significantly more concentrated in extracts after lyophilization as compared to alternative treatments (*p*-value < 0.01) (Figure 4). This result would explain the partial conversion of oleuropein to aglycone derivatives during lyophilization due to uncomplete enzymatic inactivation until a low temperature was reached.

We also monitored the formation of quinone derivatives by oxidation of secoiridoids. Oleuropein quinone was quantitatively detected at a higher concentration in extracts from leaves dried in an oven (24 h) as compared to other treatments. This was explained by the combined action between drying time and temperature since no substantial formation was observed with microwave assistance for 40 min. We also detected oleuropein aglycone quinone that was significantly enriched in extracts from leaves dried in an oven and under microwave assistance (*p*-value < 0.01). The formation of oleuropein aglycone quinone was especially favored when the sample temperature during drying was substantially increased. On the contrary, lyophilization and IAD minimized the formation of quinone derivatives from secoiridoids (Appendix A).

Contrarily to phenols, triterpenic acids were stable in extracts from leaves subjected to drying treatments with an increased temperature, which is explained by the high stability of triterpenes [30]. Therefore, IAD and MAD led to extracts with a higher content of oleanolic and maslinic acids as compared to lyophilization and oven drying. On the other hand, lyophilization and IAD are recommended treatments to obtain extracts enriched in secoiridoids with a particular variation. If the target secoiridoid is oleuropein, the recommended treatment should be IAD; if the aim is to maximize aglycone secoiridoids, the preferred treatment should be lyophilization (Appendix A).

### 3.3. Influence of Drying on the Extraction of Bioactive Compounds from Olive Pomace

In overall terms, oleacein was the most concentrated phenol in olive pomace (24.7% in Alfafara and 18.3% in Koroneiki extracts after lyophilization). This phenol was significantly more concentrated in extracts from lyophilized pomace, followed by those prepared with MAD (*p*-value < 0.01). Despite oleuropein aglycone being found at a lower concentration than oleacein in pomace extracts, the two secoiridoid derivatives reported a common pattern (Figure 5). We can explain this behavior due to the humidity. According to previous studies, the increase in relative humidity of the convention air causes a slowdown of the drying speed but a protective effect against oxidation [31]. However, MAD also promoted the oxidation of oleacein to oleaceinic acid. We found a significantly higher concentration of oleaceinic acid in extracts obtained after MAD as compared to the other strategies (*p*-value < 0.01) (Appendix A). On the other hand, this process was minimized in oven drying.

Hydroxytyrosol glucoside was also enriched in pomace extracts, which could be explained by the hydrolysis of secoiridoids that releases simple phenolic structures. Olive pomace is a residue obtained after fruit milling, malaxation, and centrifugation. The enzymatic activity is advanced as compared to fruit, and thus, cell structures are hydrolyzed. In addition, pomace humidity is particularly higher than that of leaves. Hydroxytyrosol glucoside presented a similar pattern to oleacein in samples from both cultivars, being especially enriched in extracts obtained after lyophilization and MAD (Appendix A). The result reported by MAD is explained by the high humidity content of pomace that contributes to transmitting microwave energy with high efficiency, which leads to a higher diffusion of phenolic compounds to the extraction solvent. Accordingly, hydroxytyrosol was highly concentrated in pomace extracts obtained after oven drying in Alfafara, whereas MAD led to the most concentrated extracts in Koroneiki. This could be explained by the hydrolysis of conjugated forms to release hydroxytyrosol during the long drying time (24 h) in the case of Alfafara, and during microwave irradiation for Koroneiki (Appendix A).

Thus, lyophilization would be the recommended drying treatment for pomace to obtain extracts enriched in secoiridoids, particularly in oleuropein aglycone and oleacein. On the other hand, MAD and lyophilization are the preferred strategies to obtain extracts with high contents of hydroxytyrosol glucoside, while conventional oven drying and MAD favor the isolation of hydroxytyrosol-enriched extracts, with a cultivar dependence. Concerning triterpenic acids, IAD was the suited technique to enhance the isolation of these compounds in Alfafara pomace, while lyophilization reported the highest efficiency in Koroneiki (Appendix A).

### 3.4. Influence of Drying on the Extraction of Bioactive Compounds from Olive Fruits

Olive fruits are not traditionally used for the isolation of bioactive components, as olive oil is a high-value product. However, they can provide extracts enriched in bioactive compounds with a different profile to those obtained from pomace. Oleuropein was the most concentrated phenol in fruit extracts, and this phenol was mainly isolated in extracts obtained after MAD. A derivative with an additional glucoside unit was preferentially obtained in extracts after MAD in Alfafara and after IAD in Koroneiki. Irradiation in both cases favored the rupture of plant cells to release oleuropein and oleuropein glucoside. Oleacein and oleuropein aglycone followed contrary patterns. Oleuropein aglycone was better extracted from oven-dried fruits, followed by lyophilized fruits, while IAD and MAD reported minimal concentrations of this phenol. Meanwhile, oleacein was preserved in lyophilized fruits, which can be explained by the high reactivity of this phenol with a dialdehydic structure (Appendix A).

Fruit extracts were also enriched in hydroxytyrosol glucoside and hydroxytyrosol under specific conditions. The glucoside conjugate was more enriched in extracts obtained after IAD or MAD, with a fruit size effect. Alfafara fruits are particularly bigger than Koroneiki fruits. Thus, MAD provided extracts with a higher content of hydroxytyrosol glucoside for Alfafara, while IAD provided extracts more enriched in hydroxytyrosol glucoside for Koroneiki. On the other hand, an extract enriched in hydroxytyrosol was only obtained in oven-dried fruits after a long processing time (Figure 6).

Triterpenes’ isolation from olive fruits was significantly affected by the drying procedure (*p*-value < 0.05). Notably, oleanolic and maslinic acids were enriched in extracts from fruits treated by IAD and MAD. This could be explained by the fruit structure being less compatible with oven drying and lyophilization (Appendix A).

In general terms, MAD would be suited to obtain fruit extracts enriched in oleuropein, hydroxytyrosol glucoside, and triterpenic acids. Oven drying, despite being the longest procedure, was desirable to obtain extracts with a high content of oleuropein aglycone and hydroxytyrosol. IAD would lead to extracts with a remarkable content of triterpenic acids and also hydroxytyrosol glucoside from small sized olive fruits. Finally, lyophilization was the only procedure that preserved the content of oleacein in the two cultivars.

## 4. Conclusions

This research confirms that the drying process influences the qualitative and quantitative phenolic content in extracts from leaves, fruits, and pomace. IAD was the most suited approach to obtain extracts enriched in oleuropein from olive leaves. In the case of olive pomace, lyophilization and microwave-assisted drying led to extracts concentrated in oleacein and oleuropein aglycone, two dominating secoiridoid derivatives, whereas lyophilization and MAD led to extracts with enhanced contents of hydroxytyrosol glucoside. Hydroxytyrosol was more concentrated after oven drying in Alfafara and after MAD in Koroneiki.

Oleuropein was the most concentrated phenol in fruit extracts, and this phenol was mainly found in extracts obtained after MAD. Oleuropein aglycone was better extracted from oven-dried fruits, followed by lyophilized fruits, while oleacein was preserved in lyophilized fruits. Fruit extracts were also enriched in hydroxytyrosol glucoside and after IAD or MAD.

Regarding triterpenic acids, maslinic acid and oleanolic acid reported a similar behavior between raw materials and cultivars. Thus, obtaining an extract enriched in triterpenic acids is supported by a previous drying using IAD and MAD.

## Figures and Tables

**Figure 1 foods-12-02684-f001:**
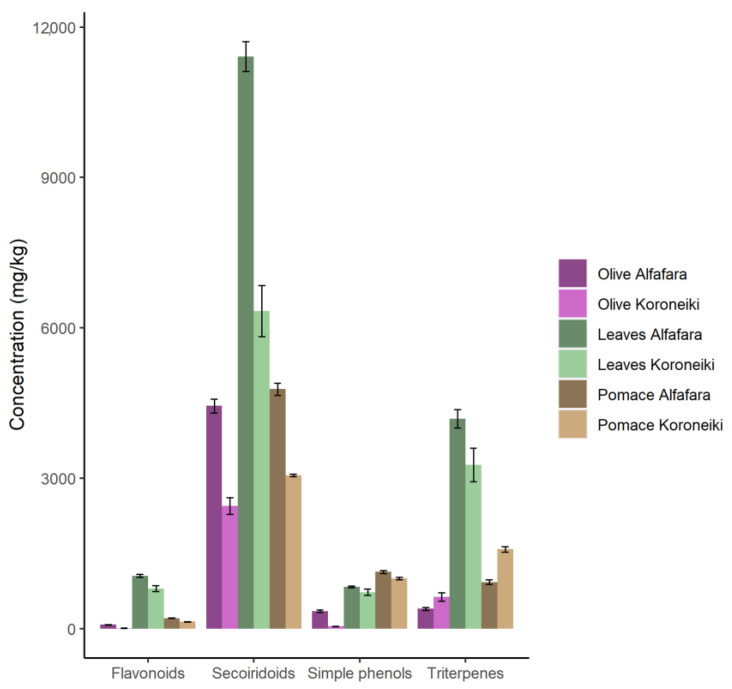
Relative concentration of main bioactive families of compounds in olive samples after lyophilization.

**Figure 2 foods-12-02684-f002:**
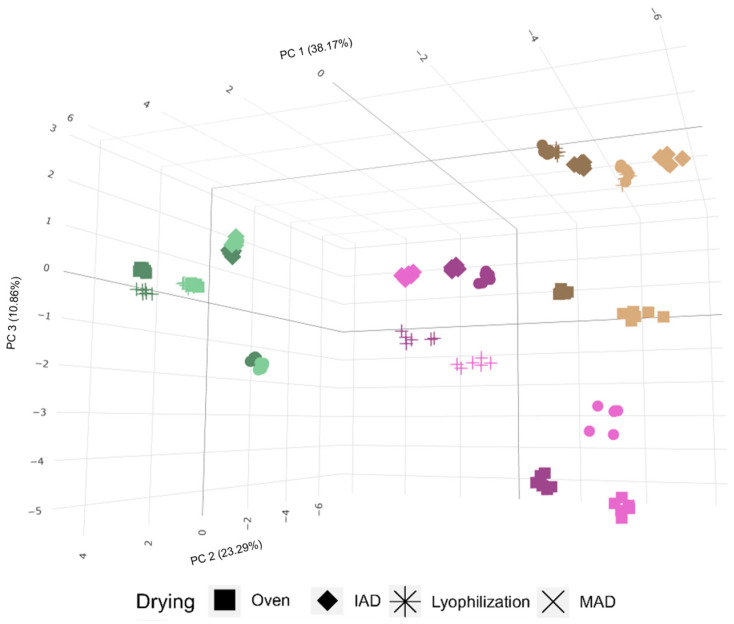
Three-dimensional PCA scores plot showing the effect of the sample as the main variability source; leaves (green), fruit (purple), and pomace (brown).

**Figure 3 foods-12-02684-f003:**
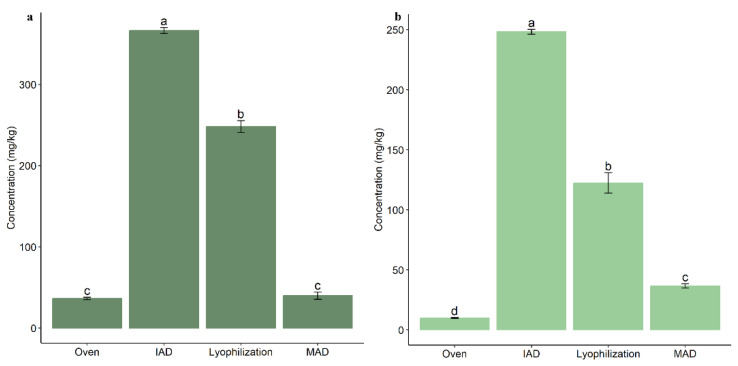
Bar plots comparing the content of oleuropein in extracts from olive leaves after application of different drying techniques: (**a**) Alfafara; (**b**) Koroneiki. Level of significance expressed as “a”, “b”, “c”, and “d” was determined by Kruskal–Wallis test with pairwise Wilcox analysis.

**Figure 4 foods-12-02684-f004:**
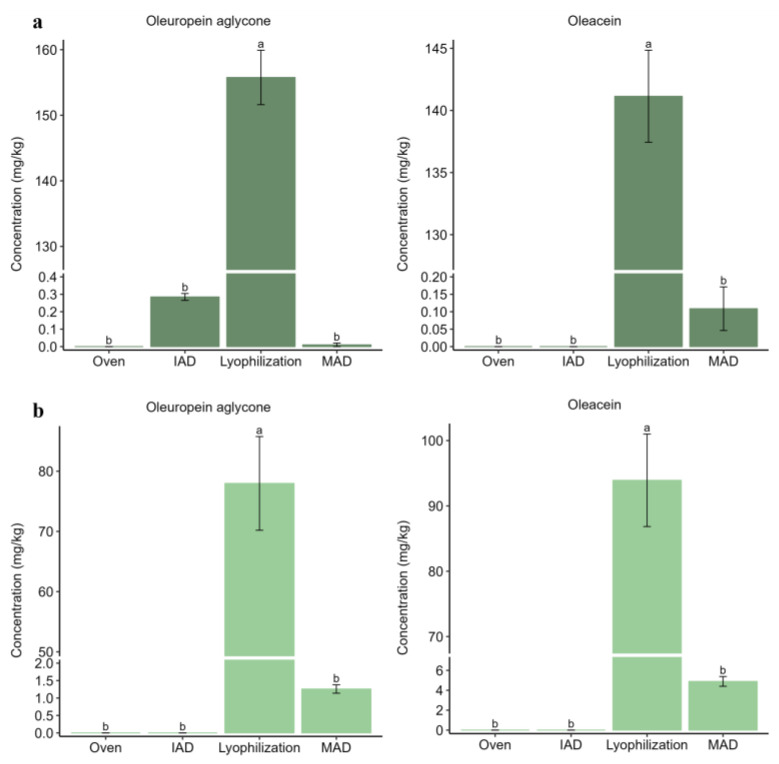
Bar plots comparing the content of oleuropein aglycone and oleacein in extracts from olive leaves after application of different drying techniques: (**a**) Alfafara; (**b**) Koroneiki. Level of significance expressed as “a” and “b” was determined by Kruskal–Wallis test with pairwise Wilcox analysis.

**Figure 5 foods-12-02684-f005:**
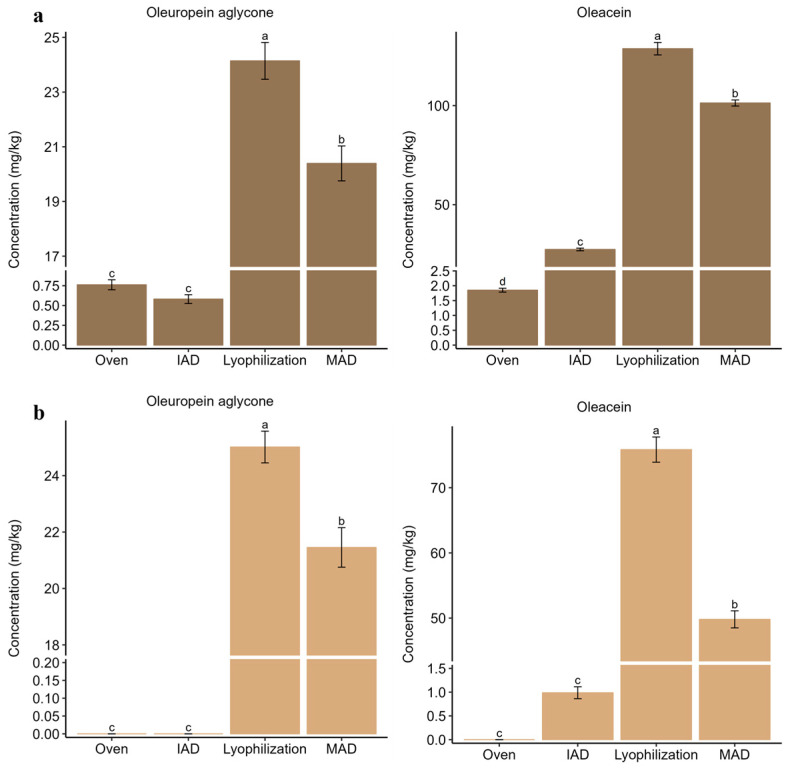
Bar plots comparing the content of oleuropein aglycone and oleacein in extracts from olive pomace after application of different drying techniques: (**a**) Alfafara; (**b**) Koroneiki. Level of significance expressed as “a”, “b”, “c”, and “d” was determined by Kruskal–Wallis test with pairwise Wilcox analysis.

**Figure 6 foods-12-02684-f006:**
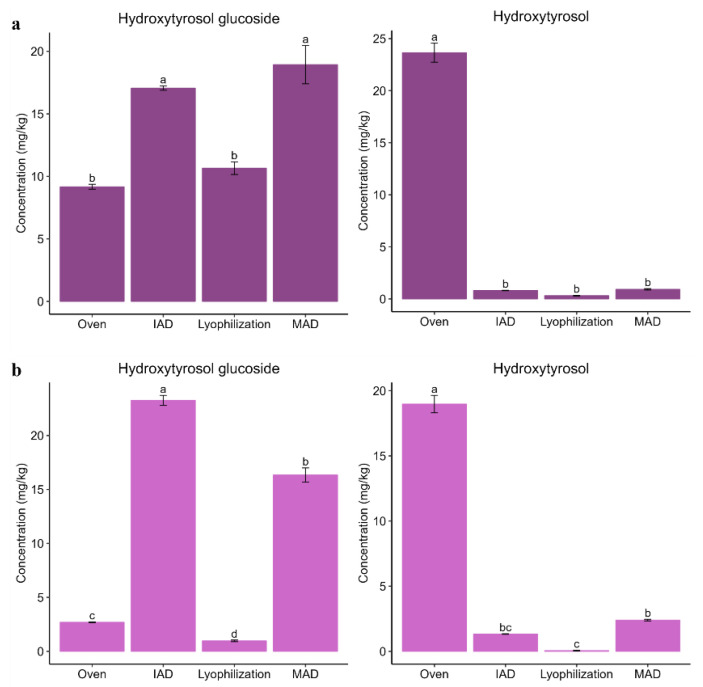
Bar plots comparing the content of hydroxytyrosol glucoside and hydroxytyrosol in extracts from olive fruits after application of different drying techniques: (**a**) Alfafara; (**b**) Koroneiki. Level of significance expressed as “a”, “b”, “c”, and “d” was determined by Kruskal–Wallis test with pairwise Wilcox analysis.

**Table 1 foods-12-02684-t001:** Summary of the conditions applied for drying olive samples by application of the different techniques (% residual humidity).

	Leaves	Pomace	Fruit
**Lyophilization**	24 h	48 h	48 h
	Under vacuum	Under vacuum	Under vacuum
	(1.70%)	(1.60%)	(2.00%)
**Oven-drying**	24 h	48 h	48 h
	45 °C	45 °C	45 °C
	(3.30%)	(3.80%)	(3.80%)
**MAD**	40 min	50 min	50–80 min
	90 W	90 W	90 W
	(2.10%)	(2.50%)	(3.20%)
**IAD**	2 h	12 h	24 h
	60 °C	60 °C	60 °C
	(2.00%)	(2.50%)	(3.30%)

## Data Availability

The data presented in this study are available in this article and in the Appendix A.

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
