# Peer review of "Comparison of Drying Techniques for Extraction of Bioactive Compounds from Olive-Tree Materials"

_foods, 2023, doi:10.3390/foods12142684_

Round 1
Reviewer 1 Report
Minor grammatical corrections can be made.
Recommendation: Major
The manuscript Comparison of drying techniques for extraction of bioactive compounds from olive-tree materials, the methodology was reasonable and technically sound.
Comments to the Author:
The main procedure and findings of the study are well expressed. Introduction: A brief survey of existing literature, the purpose, importance, and innovation of the research is well mentioned. There are major recommendations below
Point 1. The method used in PCA analysis should be better explained.
Point 2. For Figure 2, the resolution is not understood because it is poor.
Point 3. There is no need for additional files in the Foods journal. My advice is that additional tables and figures can be added to the manuscript.
Point 4. It seems that the authors did not use the 2022 and 2023 papers for weighing. I encourage them to discuss current articles.
Point 5. Add numerical results for the summary part. In the last sentence, explain your suggestion obtained by comparing drying techniques for the extraction of bioactive compounds from elderberry wood materials for future studies.
Reviewer 2 Report
The manuscript presented by Castillo-luna et al. investigates the effects of drying methods on the recovery of bioactive compounds from by-products of olive oil production and the fruit of the olive tree itself. The authors evaluated 4 drying techniques, and the results found are interesting, showing that the drying methods have a direct influence on obtaining certain classes of compounds, and that the results are also dependent on the cultivars. In general, the article is very well written with clear language, and I believe that the research carried out by the authors has scientific relevance and may be important for the development of future works and research.
I leave some minor revisions that can help in the improvement of the manuscript:
· It is interesting to include the final moisture of the samples at the end of each extraction process.
· The size and resolution of Figure 2 needs to be improved. I also suggest not using similar colors to differentiate the two cultivars. And this figure doesn't say much, but data could be included in the analysis.
· In figure 4-5, I suggest that supplementary graphs be made, with a smaller scale, to make it possible to visualize the results obtained with oven, IAD and MAD
Round 2
Reviewer 1 Report
The authors made the necessary revisions.
There are minor grammatical errors.